# Psychological Distress in South African Healthcare Workers Early in the COVID-19 Pandemic: An Analysis of Associations and Mitigating Factors

**DOI:** 10.3390/ijerph19159722

**Published:** 2022-08-07

**Authors:** Hsin-Ling Lee, Kerry S. Wilson, Colleen Bernstein, Nisha Naicker, Annalee Yassi, Jerry M. Spiegel

**Affiliations:** 1Global Health Research Program, School of Population and Public Health, University of British Columbia, Vancouver, BC V6T 1Z3, Canada; 2National Institute for Occupational Health, Division of the National Health Laboratory Service, Johannesburg 2000, South Africa; 3School of Public Health, University of the Witwatersrand, Johannesburg 2050, South Africa; 4Department of Psychology, University of the Witwatersrand, Johannesburg 2050, South Africa; 5Department of Environmental Health, Faculty of Health Sciences, University of Johannesburg, Johannesburg 2000, South Africa

**Keywords:** psychological distress, COVID-19, health care workers, risk perception, workplace management, training, practice, job stress

## Abstract

While the global COVID-19 pandemic has been widely acknowledged to affect the mental health of health care workers (HCWs), attention to measures that protect those on the front lines of health outbreak response has been limited. In this cross-sectional study, we examine workplace contextual factors associated with how psychological distress was experienced in a South African setting where a severe first wave was being experienced with the objective of identifying factors that can protect against HCWs experiencing negative impacts. Consistent with mounting literature on mental health effects, we found a high degree of psychological distress (57.4% above the General Health Questionnaire cut-off value) and a strong association between perceived risks associated with the presence of COVID-19 in the healthcare workplace and psychological distress (adjusted OR = 2.35, *p* < 0.01). Our research indicates that both training (adjusted OR 0.41, 95% CI 0.21–0.81) and the reported presence of supportive workplace relationships (adjusted OR 0.52, 95% CI 0.27–0.97) were associated with positive outcomes. This evidence that workplace resilience can be reinforced to better prepare for the onset of similar outbreaks in the future suggests that pursuit of further research into specific interventions to improve resilience is well merited.

## 1. Introduction

Following its initial identification in December 2019, the novel severe acute respiratory syndrome coronavirus 2 (SARS-CoV-2) spread rapidly and COVID-19 was declared a global pandemic by the World Health Organization (WHO) in March 2020 [1]. By August 2020, over 500,000 cases were reported in South Africa [2]. The first COVID-19 case was diagnosed in South Africa on 5 March 2020, and only 116 cases were registered by 18 March [3]. Acknowledging the rapid global spread, a National State of Disaster was declared on 15 March 2020, with schools closing on March 18 and a country-wide lockdown on 26 March 2020 [4].

As health outbreaks first emerge, healthcare workers (HCWs) worldwide are recognized to be at the forefront of screening, testing, and caring for patients, as previously observed with regard to Severe Acute Respiratory Syndrome (SARS), H1N1, Middle East Respiratory Syndrome, and other outbreaks. Subsequent to COVID-19′s emergence, HCWs in direct contact with confirmed and suspected coronavirus cases were quickly acknowledged to be vulnerable to a high risk of infection and mental health problems [5,6,7,8]. A recent scoping review of studies from various countries shows that depression, anxiety, post-traumatic stress, somatization, insomnia, and other mental health conditions were consistently prevalent among HCWs exposed to COVID-19 and other outbreaks globally [9] and specifically in South Africa [10,11].

The WHO defines mental health (MH) as “a state of well-being in which the individual realizes his or her own abilities, can cope with the normal stresses of life, can work productively and fruitfully, and is able to make a contribution to his or her community” [12]. Stress is therefore seen as a reaction to change, where someone feels overwhelmed or unable to cope. There are enormous health and cost implications at individual, organizational and societal levels when individuals are exposed to excessive stressors in acute and or chronic work stress events [13,14]. At an individual level, work stress has been shown to lead to decrements in a wide variety of health indicators, ranging from impaired psychological well-being (anxiety, depression, psychological distress, exhaustion, and burnout), physiological impairments (increases in somatic symptoms such as headaches, muscular-skeletal problems, high blood pressure, and cardiovascular disease) to an increase in negative health behaviours (increased smoking, substance abuse, and a greater dependency on prescription medications) [14]. In line with this, psychological distress has, in particular, been targeted as a key concept in appreciating [15] the emotional suffering resulting from exposure to a stressful event [16].

Subsequent to the COVID-19 pandemic spread, studies on HCWs’ knowledge, attitudes, and practices about COVID-19 have reported that HCWs were generally knowledgeable and had good practices in infection prevention [17,18,19,20,21,22] and indicate that levels of COVID-19 knowledge influenced HCWs’ attitudes or practices concerning COVID-19 [18,21,22]. Some factors identified as affecting HCWs’ knowledge of COVID-19 include training in infection prevention and access to infection prevention guidelines [18,23]. Factors such as training, lack of protective equipment (PPE) [18], and limited infection prevention and control (IPC) material [21] might also influence the prevention practices of COVID-19. These findings echo observations following the SARS epidemic indicating that direct care with SARS patients, perceived personal risks, health fear, and social isolation significantly affected HCWs’ mental health [24,25]. During the COVID-19 pandemic, while studies have investigated the impact of perceived risks, attitudes, and behaviours on physical and mental health in the general population [26] or HCWs [27], a particular gap has persisted in identifying effective measures for building resilience [28]. In fact, it has been more generally noted that greater attention is needed to broader social determinants of occupational health that impact the effectiveness of prevention and control of work-related risk [29].

In this study, we sought to identify factors and processes that contribute to the protection of HCWs from psychological distress in the context of COVID-19, and in so doing, to inform planning for future outbreaks.

## 2. Materials and Methods

### 2.1. Study Design, Participants, and Setting

We conducted a cross-sectional study within a tertiary care facility experiencing the impact of COVID-19 during a severe initial wave. In line with the multi-dimensionality involved in examining protection from psychological distress in an institution within the healthcare sector, we assessed psychological wellness, work stress, perceived risks and barriers, training, self-reported compliance with protective practices, perceived workplace hazards, and potential support relationships, including consideration of mediating and moderating factors that may be at play.

The study took place from August to October 2020 at Dr. George Mukhari Academic Hospital, situated in the north of Pretoria near the township of Ga-Rankuwa and managed by the Gauteng Department of Health. During the study period, the hospital had around 5000 employees and was a referral centre for treating critical COVID-19 patients. Initially, each facility’s Wellness Coordinator invited all staff providing services in the hospital to participate in the study online and sent out reminders. Later, the coordinators also sent out paper-based questionnaires to staff to increase the response rate and returned them for data capturing; duplicate responses were minimized by collecting names. The HCWs who provided consent were enrolled in the study. Although a diversity of languages was present in the study population, questionnaires were administered in English, the common language used in the workplace. Ethics approval for the study was obtained from the University of the Witwatersrand Human Research Ethics Committee and the University of British Columbia.

### 2.2. Data Collection

The survey encompassed three main dimensions: (1) Demographic information; (2) assessment of stress and psychological effects; and (3) perception, knowledge, attitudes, and behaviour concerning COVID-19 risks and IPC. Several questionnaires were used to collect participants’ responses on training received, perception of risks associated with COVID-19, work stress, psychological well-being, source and level of social support, knowledge, attitudes, and behaviour related to COVID-19 risks and infection control. An information sheet explaining the study, consent, and self-administered questionnaires was provided to the participants electronically. The data were collected and managed using the Research Electronic Data Capture (REDCap) tool hosted at the National Institute for Occupational Health (NIOH) [30].

### 2.3. Variables and Conceptual Model

The dependent variable of primary concern in this study was participants’ psychological well-being with a focus on the presence of psychological distress (PD). Independent variables considered in relation to this included sociodemographic data, training received, perception of risks associated with COVID-19, work stress, source and level of social support, perceived workplace hazards and workplace relationships, and self-reported compliance with protective practices when taking care of general or COVID-19 patients.

By applying a systems approach [31] to the growing scholarly literature on the effects of COVID-19 on HCWs’ mental health, the conceptual framework central to our investigation (Figure 1) is that the experience of PD at a personal (micro) level is directly and indirectly associated with the presence of job-related stresses, occupational health and safety risk management systems, and support systems (such as training or personal relationships) at an organizational (meso) level that occur in distinct national and local macro-level contexts.

### 2.4. Instruments

#### 2.4.1. General Health Questionnaire

We measured psychological well-being and related PD through the General Health Questionnaire (GHQ-12), a psychometric instrument that is widely used to screen psychological wellness and is regarded as helpful in understanding sources of distress for workers and predisposing factors [32]. The reliability coefficients have ranged from 0.79 to 0.87 in various studies [33,34,35]. A version consisting of 12 items with four answer options was used [36]. Among 12 items, the three-factor loading model proposed by Graetz is commonly mentioned [37], encompassing three dimensions of Anxiety/Depression (constantly under strain, feeling unhappy and depressed, lost sleep over worry, could not overcome difficulties), Social Dysfunction (feeling reasonably happy, playing a useful part, capable of making decisions, able to face problems, able to enjoy day-to-day activities, able to concentrate), and Confidence (losing confidence, thinking of self as worthless). Each item scores from 1 to 4 (from option 1: “better than usual” to option 4: “much worse than usual”). We applied this and then recoded the scores as 0 (if options 1 or 2) or 1 (if options 3 and 4) to calculate a total score from 0 to 12. The cut-off point for the general population was 3. The individual with a score equal to or greater than 3 is considered “psychiatric caseness” indicative of psychological distress or potential psychological problems [32,38,39].

#### 2.4.2. Job-Related Tension Index

We used the self-reporting instrument, developed initially by Kahn et al. [40], the Job-Related Tension Index (JRTI) to measure the perceptions of work stress since the beginning of the pandemic in this study. The JRTI includes 15 items that ask about the frequency of stressful experiences in the workplace, with Likert scale scores ranging from 1 to 5, with 1 (never), 2 (rarely), 3 (sometimes), 4 (often), and 5 (nearly all of the time). The questions mainly address job performance, workload, organizational design, and decision-making issues [41]. Higher mean scores indicate a higher perceived level of job-related tension. This instrument has been found valid and reliable in various organizations and industries [41,42,43,44], including healthcare [45,46,47,48]. The higher the overall score, the higher the participant’s perceived level of job-related tension.

#### 2.4.3. Perception of Risks Associated with COVID-19

We designed nine questions to measure participants’ perceived risk of COVID-19 during work, extra stress caused by work, fear of being infected or dying of COVID-19 or infecting family or others, the possibility of leaving the job, and altruistic acceptance of COVID-19 risks [49]. Each question was scored from 1 (strongly disagree) to 6 (strongly agree). The higher the scores the questions obtain, the higher the risk the participants perceive.

#### 2.4.4. Training-Related Questionnaire

We also designed the questionnaire to measure the extent of training relevant to COVID-19 IPC and PPE use. The questions asked about the participant’s training experience in hand hygiene, PPE donning and doffing, general infection control, COVID-19 IPC, ventilation requirements, and emergency procedures (e.g., evacuation). Each item is scored on a 3-point Likert scale ranging from 1 (I have not received training), 2 (I have received training), and 3 (I would like more training).

#### 2.4.5. Support

Social support is measured on a scale developed by House that initially investigated perceived social support from three sources: supervisors, colleagues, and family [50]. Our adapted scale consists of five items assessing support appreciation from each of the five sources: co-workers, supervisor, family, friends, and spouse/partner. Each item is scored on a 4-point Likert scale ranging from “not at all” (1) to “extremely” (4).

#### 2.4.6. Questionnaire on Knowledge, Attitude, Practices, and Behaviour

This questionnaire measures knowledge of the risk and concern about workplace hazards, compliance with protective practices in taking care of COVID-19 or non-COVID-19 patients, workplace relationships (participative, committed, and trust relationships), and perceptions of barriers to infection control practices relevant to COVID-19. These questions are a combination of internally designed behavioural questions and perception questions used by Wu et al. to explore the psychological impact of the SARS-CoV-1 [49]. Most of the questions are based on a Likert scale score. One question measuring the change of behaviours related to infection control practice since the COVID-19 pandemic was dichotomous (yes/no).

### 2.5. Data Analysis

We performed data analyses using the SPSS 28.0 version statistical software (IBM, Armonk, NY, USA). Descriptive analysis was performed by calculating the means or frequency of each variable against the presence of psychological distress. The mean score of each item and the average score of the questionnaire were presented as mean (M) with standard deviation (SD). Bivariate analyses were performed, including the chi-square test, Fisher’s exact test, Student’s t-test, or Mann–Whitney U test for independent variables, depending on the variable type. Hedges’ g was used to calculate the effect size (g) [51]. The suggested interpretation is that the small effect = 0.2, the medium effect = 0.5, and the large effect = 0.8 [52]. We performed correlation analysis and then logistic regression analyses, including variables or the mean score of a set of items (i.e., JRTI, Perception) with a *p*-value < 0.05, to study the predictive ability for psychological distress. Adjusted odds ratios (ORs) were calculated with a 95% confidence interval (CI). We also evaluated moderation mediation effects among variables with high correlations by using psychological distress as the dependent variable. PROCESS models (v4.0) by Andrew F. Hayes for SPSS were used to examine moderation (model 1) and mediation (model 4) effects [53]. The analysis was performed using logistic regression, calculating coefficients and standard errors using 5000 boot-strapped samples. The direct, indirect, and interaction effects were considered statistically significant when a *p*-value < 0.05 or the 95% CI does not include zero. The original questionnaires were not checked for consistency or reliability in advance. We ran a Cronbach’s alpha for internal consistency and reliability on each questionnaire after data collection. All alpha values were between 0.775 and 0.908 (acceptable to excellent), except for the questionnaire for compliance with protective practices–COVID-19, with a value of 0.662 (questionable).

## 3. Results

### 3.1. Sample Characteristics

A total of 154 HCWs participated in the survey. Among them, 122 participants (79.7%) answered all the GHQ-12 questions with a total score calculated to serve as the basis for an integrated analysis of psychological distress. Table 1 presents the characteristics of survey respondents (original 154 vs. final analysis group 122). We can see that the original and final groups’ demographic distribution patterns are highly similar. In the final group, most participants were female (90/106, 84.9%), nurses (81/120, 67.5%), and worked with admitted patients (89/95, 93.7%). The median age was 34.0 years old (Q1, Q3: 34.0, 44.0), and the median years of work experience was 8.0 years (Q1, Q3: 2.0, 12.0). Tswana is the most commonly used home language (30.1%), followed by Northern Sotho (24.8%), and Tsonga (11.5%). English as a home language was only 2.7% (3/113). Twelve participants (10.6%) mentioned they had had a COVID-19 infection before (12/113, 10.6%).

### 3.2. Psychological Distress

Table 2 provides a summary of participants’ psychological distress (PD) scores, revealing details for each GHQ-12 question. The strongest effects were for questions related to having “felt constantly under strain” (2.57 ± 1.10; 50.0% indicating worsening circumstances), “feeling unhappy or depressed” (2.46 ± 1.18; 49.0% worsened), and “enjoy[ing] … normal day-to-day activities” (2.41 ± 1.00; 42.2% worsened). In contrast, the items with the lowest scores of negative effects related to “thinking of yourself as a worthless person” (1.90 ± 1.13; 29.3% worsened), “losing confidence in yourself” (1.89 ± 1.10; 27.0% worsened), and feelings about “playing a useful part in things” (1.91 ± 0.93; 23.3% worsened).

Not all participants completed the 12 questions. The average scales for a different proportion of completed questions were compared (4-point scale). The mean values were similar either in ≥6, ≥10, or 12 questions completed (Table 2) for the 122 study participants. When scores were converted to a 2-point scale where a score of 1 was assigned to those expressing worsening circumstances and then aggregated for the 12 questions, the average score was 3.72 ± 3.23, with 57.4% of the 122 study participants presenting psychological distress, which is conventionally associated with a cut-off point of ≥3. There were no significant differences statistically in sociodemographic variables (Table 1) between the participants with (cut-off ≥ 3) and without psychological distress (cut-off < 3).

### 3.3. Work-Related Stress and Psychological Distress

Overall, there was a strong association (*p* = 0.006) between the reporting of job-related stress and PD (Table 3), with the JRTI total score for HCWs with PD (40.39 ± 12.08) higher than those without (33.45 ± 11.77). The most prevalent specific reported sources of tension (“too heavy workload” and “amount of work to do may interfere with how well it gets done”) also had the highest mean score (3.07 ± 1.36, 2.81 ± 1.24, respectively) for both groups, with a significant correlation with psychological distress. On the contrary, item 6 (Feeling not fully qualified to handle your job) had the lowest mean score (1.77 ± 1.14). Statistically, significant differences were evident for most specific items (items 1, 4, 5, 6, 9, 10, 13, 14, and 15).

### 3.4. Perception of Risks Associated with COVID-19 and Psychological Distress

Further to the relationship that we observed between general job stresses and PD (Table 3), Table 4 shows associations with specific perceptions of COVID-19 risk. While all HCWs indicated levels of concern about the consequences of exposure, it is noticeable that those who presented with PD were more likely to feel extra stress at work (4.45 ± 1.47 vs. 3.18 ± 1.49, g 0.852; *p* < 0.001) and thought about resigning because of COVID-19 (2.93 ± 1.90 vs. 1.75 ± 1.13, g 0.726; *p* < 0.001), although both groups had comparatively low mean scores regarding the thought of resigning. Altruistic acceptance of work-related risk seemed unable to lower PD, with slightly higher mean scores for those with PD (4.33 ± 1.46 vs. 3.73 ± 1.72, g 0.377). Overall, the average score significantly correlated with PD presence (4.32 ± 1.0 vs. 3.58 ± 1.06, *p* < 0.001, g 0.713).

### 3.5. Knowledge, Attitude, Practice and Behaviours during the COVID-19 Pandemic and Psychological Distress

While all HCWs expressed concerns about work-related risks (Table 5, Section 1)., those experiencing PD deemed such risks to be significantly higher, in general as well as since the emergence of COVID-19. Nevertheless, most HCWs in both groups (with or without PD, 70.10% and 81.30%) stated their behaviours related to infection control did not change or they always maintained a high level since the COVID-19 pandemic.

Those with PD were notably less likely to be in compliance with recommended protective practices related to COVID-19 (e.g., wearing an N95, gown or apron or indicating cleaning supplies unavailable) as well as general occupational safety and health provisions (Table 5, Section 2, 3.55 ± 0.84 vs. 3.90 ± 0.75, *p* = 0.020, g = −0.438). In items related to the compliance of protective practices in general care (taking care of non-COVID-19 patients), there was a statistical difference in using a surgical mask and wearing gloves between both groups (Table 5, Section 3).

Regarding workplace relationships when taking care of COVID-19 patients (Table 5, Section 4), all items were strongly correlated to psychological distress; the participants with PD had a lower average score (3.50 ± 1.03 vs. 4.29 ± 0.86, *p* < 0.001, g = −0.815); in other words, better participative, committed, and trusting workplace relationships lower the possibility of HCWs’ psychological distress.

HCWs with PD were also more prone to attribute non-compliance by HCWs to the presence of barriers to good health and safety practice in the workplace (Table 5, Section 5). The participants with PD stated the barriers occurred more frequently in each situation (item) except item 2 (poor availability of proper equipment). The mean score is significantly higher in participants with PD (3.22 ± 0.84 vs. 2.46 ± 0.94, *p* < 0.001, g 0.858).

### 3.6. Training, Supporting and Appreciation Sources and Psychological Distress

In considering factors associated with the experience of PD, we observed (Appendix A) that those receiving and seeking training and those who would like more training for COVID-19 infection control and emergency procedures tended to have less PD than those who had not received similar training (2.16 ± 0.51 vs. 1.84 ± 0.61, *p* = 0.003; 2.08 ± 0.80 vs. 1.58 ± 0.74, *p* < 0.001, respectively). Overall, participants who received no or less training would generally experience more psychological distress (*p* = 0.009, effect size 0.488).

Regarding support and appreciation at work (Appendix A), it was only the support from the immediate supervisor/boss that served to reduce the presence of PD. Better immediate supervisor/boss support appeared to minimize PD (3.27 ± 0.75 vs. 2.78 ± 1.06, *p* = 0.003, effect size −0.521). Receiving appreciation frequently from any source appeared to have no effect on reducing the presence of PD.

### 3.7. Correlation Analysis

Appendix A presents an overview of the correlations between the significant variables associated with PD (*p* < 0.05). It is noticeable that a relatively high strength of positive association existed between perceived risks of COVID-19 and JRTI (coefficient r = 0.509, *p* < 0.01) and between workplace relationships, compliance with protective practices in both COVID-19 and Non-COVID-19 patient workplace settings (r = 0.599, 0.631, 0.519, *p* < 0.01). The correlation strengths among other variables were relatively small (r = 0.1–0.3) to medium (0.3–0.5).

### 3.8. The Risk or Protective Factors of Psychological Distress

Univariate analyses identified the following significant predictors for psychological distress with *p* < 0.05: training for COVID-19 infection control, training for emergency procedures, perceived risks, JRTI, immediate supervisor/boss support, concern about the hazardous workplace in general or since COVID-19, compliance with protective practices for COVID-19 or Non-COVID-19 patients, workplace relationship, and perceived barriers to infection control practices. We further evaluated predictors of psychological distress using multiple logistic regression analyses. The first full model includes all listed variables with *p* < 0.05 (Table 6), and the final model presents only the statistically significant predictors. The final model shows that HCWs’ perceptions of risks of COVID-19 (adjusted OR 2.35, 95% CI 1.33–4.17) and barriers to compliance with infection control practices (adjusted OR 2.16, 95% CI 1.17–3.98) are two significant predictors of psychological distress. In contrast, having more emergency procedures training (adjusted OR 0.41, 95% CI 0.21–0.81) and better workplace relationships (adjusted OR 0.52, 95% CI 0.27–0.97) are significant protective factors and less likely to be associated with psychological distress.

### 3.9. Mediation and Moderation Analyses

We further explored the intertwined relationships among the factors with at least moderate strengths based on the correlation matrix. We examined the effect of perceived risks associated with COVID-19 as a mediating variable in the relationship between some predictors and psychological distress, controlling covariates having a correlation strength r ≥ 0.3 with the predictor and mediator. Appendix A describes the mediating effect of perceived risks between JRTI and psychological distress. It revealed perceived risks to COVID-19 increased as JRTI increased (β = 0.56, SE = 0.11, *p* < 0.01) and also demonstrates a significant indirect effect of perceived risks (Effect = 0.29, BootSE = 0.14, BootCI [0.04, 0.61]) between the association of JRTI and psychological distress, meaning that risk perception might partially mediate the effects of job-related tension on psychological distress.

We also examined the moderation effects among compliance variables and workplace relationships (Appendix A). The moderation analysis illustrates the significant interaction of compliance with protective practices for COVID-19 and Non-COVID-19 patient care and workplace relationships (β = −0.79, −0.99, SE = 0.29, 0.31, *p* = 0.007, 0.001, respectively), using compliance as a predictor and psychological distress as a dependent variable. The Likelihood ratio test further supports the significant interaction (chi-square: 7.81, 11.11, respectively; df = 1; both *p* < 0.01). Visualizing the plot (Appendix A) helps in understanding the conditional effect of compliance (practices on COVID-19 and Non-COVID-19 patient care) on psychological distress at different values (or levels) of workplace relationships. When there were good workplace relationships (score above the mean value), better compliance might be associated with less PD; in contrast, better compliance might significantly correlate with higher PD (i.e., aggravating instead of alleviating) when the workplace relationships were worse (score below the mean value).

## 4. Discussion

Our study, which revealed a relatively high proportion of HCWs suffering PD in the early pandemic of COVID-19, echoes findings elsewhere that greater perceived barriers to infection control practices, higher perceived COVID-19 risk, and work stress all significantly predicted psychological distress. Additionally, it demonstrated that HCWs had experienced more anxiety/depression symptoms than a loss of self-worth as a result of their experiences of working in the context of the COVID-19 pandemic. Importantly, though, we found that more training and good workplace relationships could protect HCWs’ mental health. Furthermore, we demonstrated that altering perceived risks associated with COVID-19 could mediate the effect of job-related tension on psychological distress. That workplace relationships could moderate the correlation between HCWs’ compliance with protective practices and psychological distress has important implications for policy and practice and points to the value of extending beyond attention merely to the appropriate technical measures to apply [54].

The literature on work stress and mental health indicates that in the event of high levels of work stress, and more particularly when such work stressors are perceived to be uncontrollable and unpredictable, as was the case with COVID-19 early in the pandemic, individuals may respond in a way in which they ‘give up and give in.’ Saleem et al. [55] and d’Ettorre et al. [56] note that the highly unpredictable and threatening work environment characterized by the sudden advent of COVID-19 produced unprecedented levels of stress in employees. This may explain why those with PD in the present study bothered less with compliance as they felt they had little control over whether they got infected or not. In this regard, they may have felt that irrespective of what they did, their behaviours would have little efficacy. Therefore, they may have responded in a manner of simply giving up in the face of a stressor that they felt they lacked the resources to cope with.

The literature also supports lower levels of trust and participation at work being associated with PD [55]. Chen et al. [57] and d’Ettorre et al. [56] note that supportive management is crucial in providing a work environment where trust and physical and psychological safety are enhanced, which, in turn, improves commitment, participation, and mental health. In this study, an association was indicated between lower levels of trust, participation, and commitment and workplace compliance with infection control and perceptions of barriers to infection control. It is thus likely that when individuals perceive that their work environment is not safe and they cannot trust management and organizational structures to protect them in times of high environmental pressures such as an emerging pandemic, more stress will be experienced. This, in turn, may be associated with lower compliance and a greater perception of barriers as indicated in this study. Conversely, by introducing interventions where workers can participate more and where their suggestions are taken into account, this may improve trust, participation, and commitment, enhance compliance, reduce negative perceptions, and improve mental health. In particular, this study indicated a strong association with levels of trust, participation, and commitment and greater compliance and a significant, albeit small, association with levels of trust, participation, and commitment with lower perceptions of barriers.

The study also indicated that only social support from the supervisor, more specifically instrumental support from the supervisor, was associated with lower levels of psychological distress. The literature notes that within the workplace, social support from work sources is of critical importance in lessening workplace demands, particularly in the context of health care and COVID-19 [58]. Some research has noted that it is the organizational leadership and the supervisor/manager, in particular, who are turned to as the main sources of support in times of unprecedented stress such as COVID-19 [59]. The organization and its leaders and supervisors are seen as the ‘change agents’ that are empowered to bring about ‘actual’ change in the workplace to improve working conditions. While support from other sources may have some benefit, these sources lack the authority to bring about change that can make a difference to reducing the environmental pressures that workers may be exposed to. In this regard, when introducing interventions, managers should act as the change agents and leads in bringing about change interventions that improve conditions for HCWs, thereby improving their working/environmental conditions and their mental health.

The coronavirus pandemic has caused a tremendous impact on the mental health of the general population [39,60,61,62], patient caregivers [63], and HCWs [7,9,64,65,66,67] across countries, with varying degrees of anxiety, depression, post-traumatic stress syndrome, psychological distress, and insomnia. In a systematic review with meta-analysis, Chen et al. reported the overall prevalence of mental health disorders in front-line HCWs, general HCWs, and the general population in African countries as 49%, 36%, and 38%, respectively [66]. Another review revealed an overall prevalence of mental illness symptoms in front-line HCWs and the general population in Spain of 42% and 19%, respectively [68]. The pooled prevalence of psychological distress and burnout reported by Batra et al. was 46.1% and 37.4% among HCWs [67]. Gómez-Salgado et al., using GHQ-12 measurement, showed an even higher level of psychological distress (72.0%) in the general population in Spain during the pandemic [39]. Our study found that approximately 57% of the studied HCWs presented with psychological distress, significantly higher than the general population. Notably, HCWs in South Africa have received assistance from the HealthCare Workers Care Network—a group of nearly 600 mental health professionals who volunteer to support HCWs during COVID-19 since June 2020 [69], which could be a critical point for addressing interventions continuously by hospital administration. Our data also showed more anxiety/depression symptoms than social dysfunction or loss of confidence. However, considering the strong correlation of Graetz’s three-factor model [70], further validation may be necessary for potential clinical implications.

Healthcare workers play a critical role in responding to pandemics. In doing so, they typically face uncertainty, high expectations with limited time and a high workload due to dramatically increased caseload, an overwhelmed healthcare system, and resource and staffing shortages. The WHO tracks the self-reported capacity of a country to detect and respond to a health emergency using the State Party Self-Assessment Annual Reporting [SPAR] and three voluntary components, collated by the WHO. The SPAR index of 2019 provided information on a country’s readiness to deal with the spread of COVID-19. The SPAR indicators for South Africa for 2019 showed three top challenges: health service provision; surveillance; and international health regulations coordination and focal point functions, while its laboratory services were assessed at 100%. South Africa underwent an early lockdown to allow time to increase the health system capacity, including the development of models to estimate peak required capacity and the training of repurposed staff. In the early pandemic, though, Moonasar et al. reported there was confusion due to multiple forecasting models affected by the lack of standardized surveillance systems [71], and delayed approval of a pandemic response strategy slowed the preparations and suppliers who were unable to deal with the demand, particularly for PPE. Corruption also appeared in the supply chain, draining valuable resources.

A midwife professional cohort study by Wright et al. reported that participants with higher JRTI scores believed their stress levels harmed patient care [46]. During the pandemic, some studies have also shown the influence of workload on mental health among HCWs. A study among Chinese nurses supporting Wuhan in the fight against the COVID-19 epidemic in the early pandemic reported that working hours per week was one risk factor of stress overload, and the stress overload positively correlated with self-rated anxiety [72]. Intensive work and heavy workload caused fatigue, discomfort, helplessness, and psychological distress [64,73,74]. Likewise, the participants in our study felt relatively high tension from the workload and were worried that such a workload may interfere with their performance, particularly those with psychological distress. Similarly, participants with psychological distress significantly perceived extra stress at work. Reasonable work schedules and hours and hospital administration addressing staffing issues and HCWs’ concerns may constitute effective measures in improving mental health outcomes.

The current study identified overall risk perceptions and perceived barriers to infection control practices as two significant predictors of psychological distress. Regarding the perception of COVID-19 related risks, associations existed in feeling extra stress at work, being afraid of falling ill or passing COVID-19 to others and having little control over the infection. Time, workload, lack of supervision, manager support, space, and under-training are thus essential barriers to be considered when considering adherence to recommended practices. The perceived risks and barriers identified in this study add further evidence to findings in other studies of factors commonly associated with adverse mental health conditions among HCWs [9,65,75]. For participants in our study, ‘peer pressure’ was also an associated risk that would require further exploration. Furthermore, although participants had felt extra stress at work and a fear of getting infected by or passing COVID-19, the thought of resignation (independently of mental health) was not strong. According to the government report, South Africa’s unemployment rate increased 1.7% between the third and fourth quartiles of 2020, trending up to 32.5% [76]. A high unemployment rate and the uncertainty brought by the pandemic might imply an unfavourable time for changing careers.

Our study also asked about altruistic acceptance of risk, which appeared not to have a buffering effect on psychological distress; HCWs with higher scores in feeling altruistic working with COVID-19 had a *higher* chance of developing psychological distress. The result was similar to a recent report [77] that found participants’ altruistic acceptance of risk had a small contribution to post-traumatic syndrome disease (PTSD) and depression. In contrast, some studies related to the psychological impact of SARS have found that altruistic acceptance of work-related risks was less associated with PTSD or depressive symptoms [49,78]. The role of altruistic acceptance of risk in associated psychological distress seems inconsistent; however, studies formulated this question with other predictors in different ways, such that correlation should be interpreted cautiously.

Some protective factors have previously been reported against adverse mental health during the pandemic, such as previous experience in a public health emergency, resilience, social support, and adequate PPE [9,79]. Our study also suggests that training mitigates distress.

Like the front-line HCWs, the hospital administrative system has been facing challenges during the pandemic, such as a shortage of materials or staffing and increased administrative or patient care duties [80]. So far, however, studies have rarely explored the role of leadership and organizational factors in mental health support for HCWs [64,81]. One study with Italian professionals in neuro-rehabilitation medicine units showed higher managerial support associated with less emotional exhaustion [82]. Iosim et al. found that high-quality supervision significantly reduces the risk of burnout in social workers [83]. A study exploring anxiety sources among HCWs disclosed one of the critical requests from HCWs was “Hear Me”–listen to and act on HCWs’ expert perspective and forefront experience, establishing channels of input and feedback [81]. In our study, workplace relationship as a significant predictive factor of psychological distress implies the importance of maintaining a safe, engaging, instructive, and transparent environment from which HCWs can benefit while dealing with outbreaks. Participating (reporting illness/injuries, making suggestions for correcting problems), committed relationships (obtaining supervision and instructions), and trustworthiness in leadership represent supportive strategies in building a safe work environment and were significantly associated with less psychological distress.

Moreover, workplace relationships moderate HCWs’ compliance with infection control practices, including properly donning and doffing PPE and wearing a well-fitted N95 mask. Our moderation model showed that having to adhere to physically and psychologically demanding protective practices could worsen psychological distress, especially in the offsetting of poor workplace relationships. Workplace resilience might then alleviate the impact of high compliance on HCWs’ mental health.

### 4.1. Strengths

Our survey offers a comprehensive overview of psychological well-being, job tension, social support, risk perception, training, and compliance with infection control practices among hospital HCWs during the early phase of the COVID-19 pandemic in South Africa, identifying specific perceived barriers to infection control and workplace relationships concerning COVID-19 patient care. Our findings provide a new contribution to the literature as few if any previous studies have examined these relationships in such depth and none have offered a conceptual model as to how these factors interact, especially in low-resource settings where compliance with recommended prevention and control practices has been observed to be low [84]. In particular, our focus on the organizational context in which HCWs experience job stresses and PD indicates the value of complementing investigation of effects at a micro level with critical examination of drivers at the meso level where effective interventions could be conducted–as well as considering macro-level influences on prevention and control of emerging risks during a pandemic. In this regard, our specific finding that trusted leadership and a participative management system with mutual communication approaches could be a critical support to front-line HCWs during difficult times is important, with the indication that such measures not only reduce psychological distress but support compliance with infection control measures.

### 4.2. Limitations

The cross-sectional observational design of this study has a limitation in that while it informs the presence of associations, a longitudinal study would be needed to establish cause-effect relationships and, preferably, an intervention study to test the effectiveness of the interventions suggested by our findings. In addition, this design has an unavoidable volunteer bias and a relatively small sample size (the data came from 122 participants) that could be underpowered to show associations among critical variables. Also, while participants all speak English, it is not the first language of many of them. Importantly, the study was performed prior to the development of COVID-19 vaccines at a time when HCWs had no protection except PPE and related infection control measures. Although this would be typical of circumstances at an early stage of an emerging pandemic, our results regarding the extent of psychological distress may not reflect the situation later in the pandemic, with progress achieved in treatment and vaccine protection, knowledge of the COVID-19 virus, and accumulated experience.

## 5. Conclusions

South Africa’s health care professionals working in a tertiary hospital experienced high levels of mental health disturbance during the early COVID-19 pandemic. A positive workplace relationship and more training on emergency procedures are factors that can mitigate adverse mental health. On the other hand, higher job tension, perceived risks, and more barriers to infection control practices would harm HCWs’ mental health. Policy and strategies with management, educational, and supporting initiatives should be implemented affirmatively to prevent psychological distress and improve mental health outcomes among HCWs. While the pandemic situation and vaccination roll-out continue to evolve, further research on the effectiveness of workplace interventions to reduce adverse mental health outcomes as well as protect against infection is urgently needed.

## Figures and Tables

**Figure 1 ijerph-19-09722-f001:**
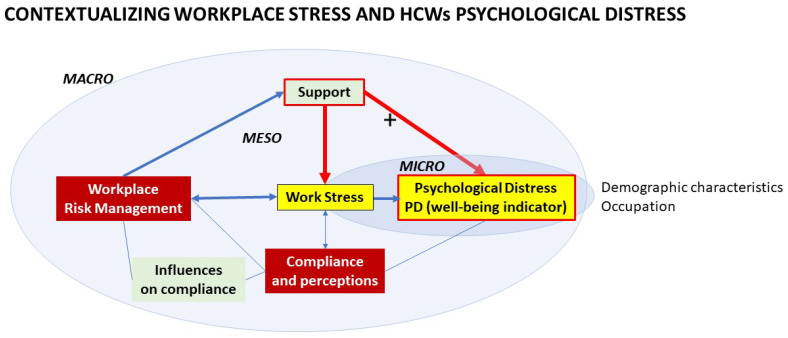
Contextualizing workplace stress and HCWs psychological distress.

**Table 1 ijerph-19-09722-t001:** Demographic characteristics of 122 participants for final analysis and comparison between the final (N = 122) and the original (N = 154) groups.

Variables	Total (N = 154) Valid/Total (%)	Final (N = 122) Valid/Total (%)
**Gender**	134/154 (87.0%)	106/122 (86.9%)
Female	116 (86.6%)	90 (84.9%)
Male	18 (13.4%)	16 (15.1%)
**Age (years)**	150/154 (97.4%)	119/122 (97.5%)
Median (IQR)	44.00 (34.00, 54.00)	34.00 (34.00, 44.00)
**Job Title**	151/154 (98.1%)	120/122 (98.4%)
Professional/Assistant/Staff nurses	100 (66.2%)	81 (67.5%)
Other professionals	51 (33.8%)	39 (32.5%)
**Work Type**	120/154 (77.9%)	95/122 (77.9%)
Inpatient	112 (93.3%)	89 (93.7%)
Outpatient only	8 (6.7%)	6 (6.3%)
**Work Years**	126/154 (81.8%)	99/122 (81.1%)
Median (IQ1, Q3)	8.00 (2.00, 12.00)	8.00 (2.00, 12.00)
**Home Language**	142/154 (92.2%)	113/122 (92.6%)
1. English	3 (2.1%)	3 (2.7%)
2. Ndebele	8 (5.6%)	8 (7.1%)
3. Northern Sotho	33 (23.2%)	28 (24.8%)
4. Southern Sotho	6 (4.2%)	2 (1.8%)
5. Swati	3 (2.1%)	3 (2.7%)
6. Tsonga	15 (10.6%)	13 (11.5%)
7. Tswana	47 (33.1%)	34 (30.1%)
8. Venda	12 (8.5%)	9 (8%)
9. Xhosa	4 (2.8%)	4 (3.5%)
10. Zulu	11 (7.7%)	9 (8%)
**Prior COVID-19 infection**	145/154 (94.2%)	113/122 (92.6%)
Yes	15 (10.3%)	12 (10.6%)
No	130 (89.7%)	101 (89.4%)
**Received flu shot this season**	148/154 (96.1%)	116/122 (95.1%)
Yes	49 (33.1%)	40 (34.5%)
No	99 (66.9%)	76 (65.5%)
**Any MH/stress management workshops prior to COVID-19**	140/154 (90.9%)	109/122 (89.3%)
Yes	53 (37.9%)	43 (39.4%)
If yes, Attended any above workshops		
Yes	36 (69.2%)	29 (69%)
No	16 (30.8%)	13 (31%)
No	87 (62.1%)	66 (60.6%)

**Table 2 ijerph-19-09722-t002:** General Health Questionnaire GHQ-12.

Item	N ^b^	Mean (SD) ^a^	% Worse ^c^
**ANXIETY/DEPRESSION ***		2.31	
5. Have you felt constantly under strain?	152	2.57 (1.10)	50.0
9. Have you been feeling unhappy or depressed?	145	2.46 (1.18)	49.0
2. Have your worries made you lose a lot of sleep?	146	2.13 (0.98)	32.1
6. Have you had the feeling that you could not overcome your difficulties?	150	2.06 (0.96)	30.6
**SOCIAL DYSFUNCTION ***		2.16	
7. Have you been able to enjoy your normal day-to-day activities?	147	2.41 (1.00)	42.2
12. Have you been feeling reasonably happy all things considered?	142	2.25 (1.00)	33.1
1. Have you been able to concentrate well on what you were doing?	151	2.14 (0.87)	28.5
8. Have you been able to face up to your problems?	148	2.11 (0.89)	29.0
4. Have you felt capable of making decisions about things?	151	2.09 (0.98)	27.2
3. Have you felt that you are playing a useful part in things?	150	1.91 (0.93)	23.3
**SELF-CONFIDENCE ***		1.89	
11. Have you been thinking of yourself as a worthless person?	147	1.90 (1.13)	29.3
10. Have you been losing confidence in yourself?	148	1.89 (1.10)	27.0
**Individual Scores (4-point scale)** ^a^	**N ^b^**	**Mean (SE)**	
Average scale (all 12 questions answered)	122	2.11 (0.06)	
Average scale (at least 10 questions answered)	147	2.16 (0.05)	
Average scale (at least 6 questions answered)	152	2.17 (0.05)	
**Individual Scores (2-point scale with a total score of 12) ^d^**	**N ^b^**	**Mean (SD) ^d^**	
Average scale (all 12 questions answered)	122	3.72 (3.23)	
**Presence of psychological distress (individual 2pt score cut point ≥ 3)**	**Yes/No**	**57.4%/42.6%**	

Note: ^a^ 4-point Likert scale (from better than usual to much worse than usual, scale from 1 to 4); ^b^ Valid responses from total n of 154 survey participants; ^c^ Responses of “worse than usual” or “much worse than usual” as % of n for category; ^d^ Responses of “worse than usual” or “much worse than usual” as % of n for category = 1. * Standardized factor loadings and between-factor correlations of Graetz’s model [32].

**Table 3 ijerph-19-09722-t003:** Association between Job-Related Tension Index and psychological distress.

Items	Psychological Distress ^a^			Effect Size
	Yes	No	*p*	g
**4. Feeling that you have too heavy a workload, one that you cannot possibly finish during an ordinary working day**	**3.34 (1.28)**	**2.71 (1.39)**	**0.010 ***	**0.476**
**13. Thinking that amount of work that you have to do may interfere with how well it gets done**	**3.09 (1.27)**	**2.43 (1.10)**	**0.004 ***	**0.545**
**5. Thinking that you will not be able to satisfy the conflicting demands of the various people over you**	**3.04 (1.17)**	**2.27 (1.06)**	**<0.001 ***	**0.678**
**1. You have too little authority to carry out the responsibilities assigned to you**	**2.99 (1.26)**	**2.37 (1.09)**	**0.007 ***	**0.514**
**9. Having to decide things that affect the lives of individuals, people that you know**	**2.94 (1.17)**	**2.39 (1.28)**	**0.015 ***	**0.450**
**15. Feeling that your job tends to interfere with your family life**	**2.91 (1.47)**	**2.20 (1.31)**	**0.006 ***	**0.507**
3. Not knowing what opportunities for advancement or promotion exist for you	2.86 (1.51)	2.39 (1.38)	0.091	
**14. Feeling that you have to do things on the job that go against your better judgement**	**2.80 (1.26)**	**2.29 (1.13)**	**0.022 ***	**0.422**
7. Not knowing what your immediate supervisor thinks of you and how he or she evaluates your performance	2.68 (1.14)	2.45 (1.27)	0.311	
11. Feeling unable to influence your immediate supervisor’s decisions that affect you	2.62 (1.23)	2.26 (1.31)	0.132	
**10. Feeling that you may not be liked or accepted by the people that you work with**	**2.58 (1.13)**	**2.08 (1.12)**	**0.016 ***	**0.444**
12. Not knowing what the people that you work with expect of you	2.44 (1.11)	2.35 (1.14)	0.639	
8. Not being able to get the necessary information to carry out your job	2.35 (1.21)	2.10 (1.20)	0.280	
2. Being unclear on just what the scope and responsibilities of your job are doing	2.18 (1.24)	2.15 (1.24)	0.879	
**6. Feeling that you are not fully qualified to handle your job**	**2.00 (1.23)**	**1.46 (0.94)**	**0.007 ***	**0.480**
**Average total score of 15 items (Mean, SD = 37.35, 12.37)**	**40.39 (12.08)**	**33.45 (11.77)**	**0.006 ***	**0.576**
**Average score of 15 items ^b^**	**2.73 (0.76)**	**2.25 (0.78)**	**0.001 ***	**0.610**

Note: ^a^ ‘Never’ to ‘Nearly all of the time’ scored on a scale from 1 to 5; * *p* < 0.05; ^b^ At least two items answered. Items with statistically significant differences are presented in bold.

**Table 4 ijerph-19-09722-t004:** Association between the perceived risks associated with COVID-19 and psychological distress.

	Psychological Distress ^a^		*p*	Effect Size
	Yes	No		g
Questions	Mean (SD)	Mean (SD)		
**3. I am afraid of falling ill with COVID-19**	**4.93 (1.30)**	**4.21 (1.52)**	**0.007 ***	**0.509**
8. My family is worried they might get COVID-19 through me	4.87 (1.50)	4.40 (1.71)	0.113	
**4. I feel I have little control over whether I get infected or not**	**4.79 (1.35)**	**4.02 (1.69)**	**0.008 ***	**0.507**
1. I believe that my job is putting me at great risk	4.54 (1.69)	4.21 (1.60)	0.287	
**2. I feel extra stress at work**	**4.45 (1.47)**	**3.18 (1.49)**	**<0.001 ***	**0.852**
**7. I am afraid I will pass COVID-19 on to others**	**4.43 (1.65)**	**3.76 (1.77)**	**0.036 ***	**0.388**
**9. I feel altruistic working with COVID-19**	**4.33 (1.46)**	**3.73 (1.72)**	**0.049 ***	**0.377**
5. I feel I may be unlikely to survive if I get COVID-19	3.47 (1.84)	2.92 (1.78)	0.101	
**6. I have thought about resigning because of COVID-19**	**2.93 (1.90)**	**1.75 (1.13)**	**<0.001 ***	**0.726**
**Average score of 9 items ^b^**	**4.32 (1.00)**	**3.58 (1.06)**	**<0.001 ***	**0.713**

Note: ^a^ ‘Strongly disagree’ to ‘Strongly agree’ scored on a scale from 1–6. * *p* < 0.05; ^b^ At least two items answered. Items with statistically significant differences are presented in bold.

**Table 5 ijerph-19-09722-t005:** Association between knowledge, attitudes, and practices towards COVID-19 and psychological distress.

	Psychological Distress		*p*	Effect Size
	Yes	No		g
Items	Mean (SD)	Mean (SD)		
**Section 1. Concern about workplace hazards (higher scores meaning less risky and more protection in the workplace)**				
1. How hazardous do you feel your workplace is in general?	1.37 (0.64)	1.76 (0.77)	0.003 *	−0.556
(Risky to No risk, score from 1 to 4)				
2. How hazardous do you feel your workplace is since the COVID-19 pandemic began?	1.22 (0.54)	1.48 (0.74)	0.038 *	−0.409
(Risky to No risk, score from 1 to 4)				
3. Has your behaviour related to infection control changed since COVID-19 pandemic?	(%)	(%)		
Yes (following the infection control protocol)	29.90	18.80	0.176	
No (Always maintained a high level of infection control/not exposed or at risk)	70.10	81.30		
4. From your perspective, is the supply of PPE available to you?	1.57 (0.70)	1.65 (0.56)	0.496	
(insufficient/adequate/sufficient, score from 1 to 3)				
**Section 2. Compliance with protective practices-COVID-19 (from ‘Never’ to ‘Always’, score 1–5)**				
1. I use eye and face protection (other than prescription glasses)	3.41 (1.38)	3.80 (1.57)	0.164	
2. I use a surgical mask when dealing with patients when an N95 is not available	4.31 (1.15)	4.10 (1.49)	0.406	
3. I wear an N95	3.22 (1.56)	4.06 (1.49)	0.004 *	−0.545
4. I reuse my disposable masks	2.61 (1.66)	2.23 (1.66)	0.234	
5. I wear a gown or apron	3.61 (1.52)	4.27 (1.19)	0.011 *	−0.471
6. I wear gloves	3.97 (1.37)	4.40 (1.20)	0.086	
7. Cleaning supplies are available	3.80 (1.29)	4.38 (1.12)	0.015 *	−0.464
Average score of 7 items ^a^	3.55 (0.84)	3.90 (0.75)	0.020 *	−0.438
**Section 3. Compliance with protective practices-non-COVID-19 (from ‘Never’ to ‘Always’, score 1 to 5)**				
1. I use a surgical mask	4.29 (1.26)	4.85 (0.55)	0.001 *	−0.539
2. I wear gloves	3.53 (1.45)	4.14 (1.50)	0.028 *	−0.414
3. I wear a gown/apron	3.66 (1.37)	4.06 (1.51)	0.143	
4. I send them for testing if they report one symptom	3.83 (1.40)	4.08 (1.30)	0.318	
5. I send them for testing if they report more than one symptom	4.29 (1.18)	4.39 (1.20)	0.660	
Average score of 5 items ^a^	3.92 (0.88)	4.31 (1.00)	0.029 *	−0.412
**Section 4. Workplace-participative, committed and trusting relationship when taking care of COVID-19 patients (from ‘Never’ to ‘Always’, score 1–5)**				
1. I make suggestions for correcting health and safety problems	3.35 (1.43)	3.88 (1.30)	0.045 *	−0.378
2. I get clear supervision regarding safe work practices	3.32 (1.37)	4.33 (1.00)	<0.001 *	−0.813
3. I am encouraged to report injuries and illness at work	3.93 (1.25)	4.63 (0.86)	<0.001 *	−0.633
4. I trust my manager to make my workplace safe	3.70 (1.31)	4.55 (0.94)	<0.001 *	−0.726
5. I trust my health and safety committee to make my workplace safe	3.12 (1.43)	4.00 (1.19)	<0.001 *	−0.653
Average score of 5 items ^a^	3.50 (1.03)	4.29 (0.86)	<0.001 *	−0.815
**Section 5. Perceptions of barriers to infection control practices (from ‘Never’ to ‘Always’, score 1–5)**				
1. Not enough time/too much work	3.74 (1.21)	2.90 (1.25)	<0.001 *	0.686
2. Poor availability of proper equipment	3.71 (1.26)	3.33 (1.45)	0.129	
3. Lack of proper space	3.61 (1.26)	2.78 (1.42)	0.001 *	0.623
4. Do not feel the risk that requires it	3.14 (1.32)	2.40 (1.28)	0.003 *	0.567
5. Peer pressure	2.83 (1.51)	1.64 (1.06)	<0.001 *	0.888
6. Not trained well enough	3.13 (1.49)	2.42 (1.33)	0.009 *	0.502
7. Proper supervision not provided	2.91 (1.34)	2.25 (1.28)	0.010 *	0.497
8. Not supported by my manager	2.67 (1.34)	2.00 (1.23)	0.007 *	0.512
Average score of 8 items ^a^	3.22 (0.84)	2.46 (0.94)	<0.001 *	0.858

Note: * *p* < 0.05; ^a^ At least two items answered.

**Table 6 ijerph-19-09722-t006:** Multiple logistic regression models on psychological distress by major variables.

	Psychological Distress (Total Score ≥ 3)	
Variables	Full Model	Final Model
	OR (95% CI)	Adjusted OR (95%CI)
**Training**		
COVID-19 infection control	0.37 (0.18, 0.74) **	
Emergency procedures	0.44 (0.27, 0.72) **	0.41 (0.21, 0.81) *
**Perceived risks of COVID-19**	1.99 (1.36, 2.93) **	2.35 (1.33, 4.17) **
**Job-Related Tension Index**	2.24 (1.35, 3.72) **	
**Social support**		
Immediate supervisor/boss	0.56 (0.37, 0.86) **	
**Concern about workplace hazards**		
Feel workplace is hazardous in general	0.46 (0.27, 0.79) **	
Feel workplace is hazardous since the pandemic	0.53 (0.29, 0.95) *	
**Compliance with protective practices (COVID-19)**	0.57 (0.35, 0.92) *	
**Compliance with protective practices (non-COVID-19)**	0.63 (0.41, 0.96) *	
**Workplace relationships**	0.40 (0.25, 0.64) **	0.52 (0.27, 0.97) *
**Perceptions of barriers to infection control practices**	2.67 (1.66, 4.29) **	2.16 (1.17, 3.98) *

Note: * *p* < 0.05; ** *p* < 0.01.

## Data Availability

De-identified participant data containing possibly identifiable data and personal medical data may be obtained on application to the National Health Laboratory.

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
