# Peer review of "Psychological Distress in South African Healthcare Workers Early in the COVID-19 Pandemic: An Analysis of Associations and Mitigating Factors"

_ijerph, 2022, doi:10.3390/ijerph19159722_

Round 1
Reviewer 1 Report
Dear Authors,
Thank you for submitting this paper. Reading and reviewing this work was an honour and an exciting opportunity.
In this paper, you presented the results of a survey with 154 participants and several instruments such as the General Health Questionnaire, Job-related Tension Index, Perception of risk associated with COVID-19, Training, Support etc., questionnaires.
The paper appropriately contains the problem statement, literature background, and methodology (instruments and data analysis). The abstract and keywords are adequate.
The ISO/PAS 45005:2020 Occupational health and safety management — General guidelines for safe working during the COVID-19 pandemic standard is a fundamental tool to combat COVID-19, so I suggest including it in the paper and use it in the future.
Please consider improving Figure 1 and use the same terminology in the text, e.g. show PD on the figure.
You should add the heading number in line 188.
Please give the origin of the Knowledge, Attitude, Practices and Behaviour Related Questioner.
In table 1, you should want to change 'sex' to 'gender'. Please check age distribution because it seems in table 1 that when you reduced the number of participants from 154, you excluded the young and elderly.
Please use a consistent and more comprehensible notation for table footnotes; you have a, *, § etc. Also, please add an explanation of Psychological Distress YES / NO values and why they change in different tables.
In line 288, please, give a more detailed comment on table 3 or delete it from the paper.
In line 324, please clarify who are "those" participants.
In line 410, correct the name as in line 417.
Please, improve the grammar in line 415.
The discussion gives a good overview of the topic and puts the results in place. I would add to the limitations that all data come from 122 answers.
The conclusion summaries and highlights the supported findings of the research.
In summary, I like this work, the presentation of results and the discussion.
Author Response
In this paper, you presented the results of a survey with 154 participants and several instruments such as the General Health Questionnaire, Job-related Tension Index, Perception of risk associated with COVID-19, Training, Support etc., questionnaires.
The paper appropriately contains the problem statement, literature background, and methodology (instruments and data analysis). The abstract and keywords are adequate.
The ISO/PAS 45005:2020 Occupational health and safety management — General guidelines for safe working during the COVID-19 pandemic standard is a fundamental tool to combat COVID-19, so I suggest including it in the paper and use it in the future.
We have added this reference to the beginning of the Discussion section [new line 420], to acknowledge how attention to workplace relationships adds to consideration of appropriate technical measures to pursue.
Please consider improving Figure 1 and use the same terminology in the text, e.g. show PD on the figure.
The text and figure have been modified, now more clearly designating the relationship with PD (psychological distress). [new lines 135-143]; marked version lines 137-146].
You should add the heading number in line 188.
Done [marked version lines 196].
Please give the origin of the Knowledge, Attitude, Practices and Behaviour Related Questioner.
Done. The citation was inserted in 2.4.6. These questions are a combination of internally designed behavioural questions and perception questions used to explore the psychological impact of the SARS-CoV-1 by Wu et al. 2009. [marked version lines 207-209].
In table 1, you should want to change 'sex' to 'gender'. Please check age distribution because it seems in table 1 that when you reduced the number of participants from 154, you excluded the young and elderly.
- 'Sex' has been revised to 'Gender' in Table 1. [marked version line 254].
- Based on the age distribution for the sets of 154 (left) vs 122 (right) respondents, the numbers of older respondents were not changed much. The numbers for respondents of other ages seemed to be unequally reduced, but the proportion of each age category was similar.
Please use a consistent and more comprehensible notation for table footnotes; you have a, *, § etc. Also, please add an explanation of Psychological Distress YES / NO values and why they change in different tables.
- The English alphabet was used for the notation in the table footnotes, except '*' for the p values.
- The individual with GHQ-12 scores equal to or greater than 3 (2 point scale, see 2.4.1) is considered "psychiatric caseness" across all the analyses and tables. We presented different average scales (using the original 4 point scale) for a different proportion of completed questions in Table 2 to show the feasibility of 122 participants' data for analysis. The authors added an explanation, as suggested. We have added wording to the line in the table presenting this finding.
In line 288, please, give a more detailed comment on table 3 or delete it from the paper.
Line 277-285 presented the results and comments in Table 3. 'Table 3' is indicated in the revised version.
In line 324, please clarify who are "those" participants.
We have replaced "those participants" with "participants with PD."
In line 410, correct the name as in line 417.
Done.
Please, improve the grammar in line 415.
Done. This section has been rewritten into three sentences ( to enhance meaning) and all have been revised grammatically.
The discussion gives a good overview of the topic and puts the results in place. I would add to the limitations that all data come from 122 answers.
Done. That the data comes from 122 participants' is now explicitly indicated.
The conclusion summaries and highlights the supported findings of the research.
In summary, I like this work, the presentation of results and the discussion.
We thank Reviewer 1 for these appreciative comments – and for the suggestions provided for clarifying and improving the paper.

Reviewer 2 Report
Thank you for further expanding the knowledge about the psychological stressors related to the pandemic of COVID-19. The detailed analysis of the questionnaires, as well as their relationships, is quite interesting. However, as in any type of research, some aspects of style and content could be improved. Here are some suggestions:
- Line 15: Please eliminate one-word "Correspondence:".
- Line 25: Throughout the manuscript, sometimes the expression of the p-value appears with unity (e.g., p<0.05) and sometimes not (p<.05). Please unify the criterion for the entire manuscript.
- Lines 30-31: If possible, it is recommended to use controlled vocabulary thesauri for keywords.
- Line 64: The closing parenthesis is missing (before the "[14]").
- Lines 67-80: Is it possible for the font colour to be dark grey instead of black?
- Line 91: If it is a cross-sectional study, why has not the STROBE Checklist for cross-sectional studies been used? (https://www.strobe-statement.org/). It would be advisable to use it and attach it as Supplementary Material to show the methodological quality of the study.
- Lines 103-105: I understand that the questionnaires were distributed in an online format at the beginning of the process. What precautions were taken to avoid duplicate responses from participants?
- Lines 145-146: The study you cite ([32]) does not specify which studies obtained these validity coefficients. Please cite some studies that have used the questionnaire and that have coefficients in that range of values.
- Line 158: The period is missing at the end of the paragraph.
- Line 161: A space should be added in "al.[37]".
- Line 188: "2.4.5" should be added to the subsection's title.
- Line 211: If the sample was larger than 20 and Cohen's d and Hedges' g are similar on this assumption. Was there any other particular reason for using g versus d?
- Lines 232-234: Why use median and interquartile ranges if all other descriptive results use mean and standard deviation?
- Table 1. Cont: There is a typo in the row "Prior Covid infection", specifically in "113/112". Moreover, although the abbreviation "MH" is understood, it has not been specified above. I suggest including the abbreviation in line 52.
- Line 250: Please add a space in "42.2%worsened".
- Line 254-256: This last sentence is more appropriate for the Discussion.
- Line 273: The title of the subsection should be italicized.
- Tables 3, 4, 5, and S1: If a result gets p>0.05, please do not add its corresponding effect size.
- Line 293: Please replace "effect size" with "g". Keep this in mind for the rest of the manuscript.
- Line 299: I don't know if the expression "p=<0.001" is correct or if it is a typo. Please use the mathematical symbol that includes both symbols if it is correct.
- Line 311: It has not been stated above what "OSH" means.
- Lines 311-312: This result does not appear in Table 5. Please indicate where it comes from. If necessary, I recommend eliminating it.
- Line 410: Remove an apostrophe in "d''Ettorre".
- Lines 438 and 440: Add the hyphen in "COVID 19".
- Lines 449-468: After reading this paragraph, there is no mention of patient caregivers. Some studies also analyze, directly or indirectly, the mental health of this profile of people (e.g. https://doi.org/10.3390/healthcare10030523). It would be useful to comment on some details about this.
- Lines 540-541: Following the same style, replace "Iosim and colleagues" with "Iosim et al.".
- Lines 569-574: In my opinion, the text of these lines is very intricate and complex. Could you rewrite it more simply?
- References: Some references have details that need to be solved, especially in punctuation marks or typos. Some references to check would be numbers 8, 12, 30, 32, 33, 47, 63 and 74.
I hope you find my recommendations helpful.
Best regards.
Author Response
- Line 15: Please eliminate one-word "Correspondence:".
The mandatory indication of where correspondence should be directed has been revised. Annalee Yassi now replaces Jerry Spiegel as the corresponding author. The initials KW have been removed and are represented by [email protected], who will be the contact person for getting access to the data set.
- Line 25: Throughout the manuscript, sometimes the expression of the p-value appears with unity (e.g., p<0.05) and sometimes not (p<.05). Please unify the criterion for the entire manuscript.
Done. P values have been unified across the tables and content with the format of 0.05 or 0.001.
- Lines 30-31: If possible, it is recommended to use controlled vocabulary thesauri for keywords.
The authors revised some keywords to match the subject terms (EBSCOhost databases).
- Line 64: The closing parenthesis is missing (before the "[14]").
Done.
- Lines 67-80: Is it possible for the font colour to be dark grey instead of black?
Done. The colour for Lines 71-87 (one paragraph) changed to grey with 50% darker.
- Line 91: If it is a cross-sectional study, why has not the STROBE Checklist for cross-sectional studies been used? (https://www.strobe-statement.org/). It would be advisable to use it and attach it as Supplementary Material to show the methodological quality of the study. [
STROBE Checklist is now included as Supplementary Material.
- Lines 103-105: I understand that the questionnaires were distributed in an online format at the beginning of the process. What precautions were taken to avoid duplicate responses from participants?
Duplicate responses were minimised by collecting names. An explanation was added to 2.1.
- Lines 145-146: The study you cite ([32]) does not specify which studies obtained these validity coefficients. Please cite some studies that have used the questionnaire and that have coefficients in that range of values.
Done. The citations have been updated.
- Line 158: The period is missing at the end of the paragraph.
Done.
- Line 161: A space should be added in "al.[37]".
Done.
- Line 188: "2.4.5" should be added to the subsection's title.
Done.
- Line 211: If the sample was larger than 20 and Cohen's d and Hedges' g are similar on this assumption. Was there any other particular reason for using g versus d?
Hedge's g considers each sample size when calculating the overall effect size. Hedge's g would be recommended when two sample sizes are not equal, which is the situation this study had (PD vs non-PD).
- Lines 232-234: Why use median and interquartile ranges if all other descriptive results use mean and standard deviation?
The distribution of age and work years are not a normal distribution. Therefore, a median with IQR was used. As to the questionnaire (e.g., perception, JRTI), scales were pooled and calculated from the mean values of the total questions from each participant (i.e., mean of the mean values), which were approaching a normal distribution, so that mean values were suitable to use.
- Table 1. Cont: There is a typo in the row "Prior Covid infection", specifically in "113/112". Moreover, although the abbreviation "MH" is understood, it has not been specified above. I suggest including the abbreviation in line 52.
Done. MH has now been added.
- Line 250: Please add a space in "42.2%worsened".
Done.
- Line 254-256: This last sentence is more appropriate for the Discussion.
Done. The sentence was moved to the first paragraph of Discussion.
- Line 273: The title of the subsection should be italicized.
Done.
- Tables 3, 4, 5, and S1: If a result gets p>0.05, please do not add its corresponding effect size.
Done.
- Line 293: Please replace "effect size" with "g". Keep this in mind for the rest of the manuscript.
Done. Hedge's g has been replaced by "g."
- Line 299: I don't know if the expression "p=<0.001" is correct or if it is a typo. Please use the mathematical symbol that includes both symbols if it is correct.
Done. The typo "=" was deleted.
- Line 311: It has not been stated above what "OSH" means.
OSH was replaced by "occupational safety and health."
- Lines 311-312: This result does not appear in Table 5. Please indicate where it comes from. If necessary, I recommend eliminating it.
Done. Data were revised and indicated.
- Line 410: Remove an apostrophe in "d''Ettorre".
Done.
- Lines 438 and 440: Add the hyphen in "COVID 19".
Done.
- Lines 449-468: After reading this paragraph, there is no mention of patient caregivers. Some studies also analyze, directly or indirectly, the mental health of this profile of people (e.g. https://doi.org/10.3390/healthcare10030523). It would be useful to comment on some details about this.
We have restructured the sentence and included reference to effects on the mental health of patient caregivers (in addition to the general population and HCWs).
- Lines 540-541: Following the same style, replace "Iosim and colleagues" with "Iosim et al.".
Done.
- Lines 569-574: In my opinion, the text of these lines is very intricate and complex. Could you rewrite it more simply?
We have edited this section so that it is clearer and flows better.
- References: Some references have details that need to be solved, especially in punctuation marks or typos. Some references to check would be numbers 8, 12, 30, 32, 33, 47, 63 and 74.
Done. All references have been rechecked.
I hope you find my recommendations helpful.
We thank Reviewer 2 for the very helpful recommendations for improving the paper.

Reviewer 3 Report
Dear Authors,
Thank you for the opportunity to review this paper. In my opinion, the manuscript Psychological Distress in South African Healthcare Workers early in the COVID-19 Pandemic: An Analysis of Associations and Mitigating Factors addresses the topic of great clinical interest. The issue raised in this manuscript is very important and necessary for public discussion. The article is generally well written, the argumentation is sound, and the paper is well structured. The statistical approach is appropriate for testing variables and the conceptual model.
The only limitation that I noticed in the methodological part was the lack of described psychometric properties of the tools used. My question is whether the authors tested the reliability of the questionnaires they used? On page 4 it says that other studies confirm this, but did the authors in these specific studies check whether these tools on their sample of people had an equally good internal consistency? I would also ask the authors to complete the information in the article on whether they checked the accuracy and reliability of the tools they created themselves.
The results and discussion are clearly presented. The references are appropriate and adequate to related and previous work.
Author Response
The only limitation that I noticed in the methodological part was the lack of described psychometric properties of the tools used. My question is whether the authors tested the reliability of the questionnaires they used? On page 4 it says that other studies confirm this, but did the authors in these specific studies check whether these tools on their sample of people had an equally good internal consistency? I would also ask the authors to complete the information in the article on whether they checked the accuracy and reliability of the tools they created themselves.
The results and discussion are clearly presented. The references are appropriate and adequate to related and previous work.
Reply:
The questionnaires were not checked for consistency or reliability in advance. Instead, we ran a Cronbach's alpha for internal consistency/reliability on each questionnaire after data collection. All alpha values were between 0.775 and 0.908 (acceptable to excellent), except the questionnaire for compliance with protective practices – COVID, with a value of 0.662 (questionable). Relevant information was added in the section on data analysis.
We thank Reviewer 3 for the very helpful recommendations for improving the paper.
